# ORTHOGONAL EVALUATIONS ENABLE MORE ROBUST PREDICTIONS OF PROTEIN-LIGAND INTERACTIONS

**Joseph G. Wakim**[1]**, Irene Kim**[1]**, Adam T. Zemla**[1,*]
[1]Computing Directorate, Lawrence Livermore National Laboratory, Livermore, CA 94550, USA
[*]Correspondence to zemla1@llnl.gov

## ABSTRACT

Computational models can predict protein-ligand interactions (PLIs) at scales that far surpass experimental validation, which makes reliable confidence estimation critical. Existing approaches use protein structure and function as complementary, independently derived comparators for predicting and evaluating PLIs. However, function-based evaluations perform poorly for promiscuous ligands, which target proteins with diverse functions. Accordingly, confidence estimation for modeled PLIs involving promiscuous ligands remains an open challenge. To address this gap, we introduce a novel physicochemical representation as an additional comparator for evaluating PLIs. Our representation encodes binding-pocket-specific features along a protein's surface, which influence its affinity for a ligand. In preliminary experiments on PLIs involving promiscuous ligands, we find that incorporating these features yields more robust confidence estimates compared to using structure and function alone. These results suggest that physicochemical representations capture meaningful biological signals for prioritizing high-quality drug leads, motivating a multimodal evaluation framework for drug discovery.

## 1 INTRODUCTION

Proteins function as molecular machines that drive biological processes. By binding to pockets on a protein's surface, small-molecule drugs can inhibit or activate the protein's function, leading to therapeutic or toxic effects. Computer-aided drug discovery (CADD) involves predicting PLIs with computational models to develop new drugs, anticipate off-target effects, or repurpose existing drugs (Sabe et al., 2021; Carpenter & Altman, 2024). The field benefits from the growing availability of protein sequences, structural data, and functional information (Boutet et al., 2007; The UniProt Consortium, 2025; The Gene Ontology Consortium et al., 2023; Varadi et al., 2022), which has accelerated model development and broadened the space of testable hypotheses. Yet, given the costs and risks associated with clinical trials, only a limited number of predictions can be experimentally verified (Martin et al., 2017). There is an open need for robust metrics that score predictions and prioritize high-quality drug leads.

In this work, we leverage independently derived properties describing protein-ligand complexes to rank predicted PLIs based on their quality. Given a protein model, we first quantify how well the structures of the protein's binding pockets align with known protein-ligand complexes (Zemla et al., 2022). Strong alignments (*i.e.*, local structural similarities) offer a preliminary indicator of an interaction. For each putative PLI derived from the structural model, we then score how similar the functions of the protein are to those of known targets for the ligand (Wakim et al., 2025). However, for promiscuous ligands that bind proteins with diverse or unrelated functions, this functional comparator can be unreliable, which limits our ability to confidently rank their predicted PLIs. To address this limitation, we introduce a novel physicochemical representation of protein binding pockets as an additional comparator for evaluating interactions. We find that incorporating this representation improves evaluations of PLIs involving promiscuous ligands. Although structure, function, and physicochemical properties are biologically linked, their observed correlations are weak, which suggests that each modality captures a distinct source of signal and uncertainty. Putative interactions that are jointly supported by structural alignment, physicochemical compatibility, and functional similarity emerge as high-confidence candidates for follow-up testing. By simultaneously evaluating these orthogonal sources of evidence, we provide a more robust scoring framework for CADD that prioritizes experimentally testable interactions and accelerates discovery.

## 2 METHODS

We independently derive human protein and binding site representations based on their structural, functional, and physicochemical properties, providing orthogonal evidence for predicting and evaluating PLIs. We use structural models to identify candidate interactions and compute functional and physicochemical "misalignment scores" to assess them. By considering these three modalities simultaneously, we can better prioritize high-quality PLIs as candidates for future drug development.

### 2.1 PROTEIN REPRESENTATIONS

**Structure.** First, using AlphaFold 2, we model the 3D configuration of human proteins (Jumper et al., 2021; Varadi et al., 2022). For each modeled protein, we use a tool called PDBspheres (Zemla et al., 2022) to detect pockets on the protein's surface that locally align with known protein-ligand complexes in the Protein Data Bank (PDB) (Burley et al., 2022); strong alignment with an experimental complex suggests that the modeled protein may bind the complex's ligand. We quantify alignment quality using the local-global alignment (LGA) metric, with greater values indicating stronger alignments (Zemla, 2003). However, even with strong LGA scores, predicted interactions are sensitive to the quality of the associated protein model.

**Function.** We leverage functional insights to verify predictions obtained from structural models. Using protein sequence (The UniProt Consortium, 2025), Gene Ontology (GO) annotations (The Gene Ontology Consortium et al., 2023), reported interactions (Zdrazil et al., 2024; Heinzke et al., 2024), and three deep learning models (Zhong et al., 2020; Ieremie et al., 2022; Sanderson et al., 2023), we form protein-function representations specialized for evaluating PLIs. Our function-based evaluation framework is described by Wakim et al. (2025). However, functional annotations and associated embeddings describe proteins in their entirety and, therefore, lack the pocket-specific resolution of our structural models. Additionally, functional properties are of limited utility for evaluating PLIs involving promiscuous ligands, which target diverse sets of proteins.

**Physicochemistry.** For more detailed evaluation of predicted interactions, we introduce **ResNum**, a curated set of 235 physicochemical features that characterizes individual protein binding pockets. These features are computed on experimentally resolved protein-ligand interfaces, or "spheres" (Zemla et al., 2022). ResNum features quantify properties like hydrophobicity, aromaticity, and sulfur content for different layers of contact residues. When residues in a protein model locally align with those from a sphere, we transfer the corresponding ResNum features to the modeled site. We compress the ResNum feature vectors using PCA, forming 47-dimensional embeddings (capturing 99% of the explained variance) that represent individual binding pockets. We obtain protein-level representations by averaging the ResNum embeddings across all binding pockets, weighting each pocket by the quality of its alignment to the corresponding reference sphere. In addition to evaluating predicted PLIs, our ResNum embeddings allow us to evaluate the predicted interaction sites.

### 2.2 EVALUATION FRAMEWORK AND DEMONSTRATION

We adapt a framework introduced by Wakim et al. (2025) to evaluate predicted PLIs based on functional and physicochemical properties. At a high level, our framework involves comparing properties of proteins implicated in the predicted interactions to those associated with known PLIs. We compute "misalignment scores" indicating how consistent predicted interactions are with reported PLIs (lower misalignment scores suggest greater consistency). These misalignment scores can be used to prioritize high-quality drug leads. Functional and physicochemical properties of reported PLIs are derived from distinct data sources; functional properties are assessed from interactions, sequences, and GO annotations reported in ChEMBL (Zdrazil et al., 2024; Heinzke et al., 2024), while physicochemical properties are computed from structural models in the PDB (Zemla et al., 2022; Burley et al., 2022). Therefore, the two properties encode distinct sources of signal and noise, and jointly considering their misalignment scores can improve the robustness of our evaluation.

To demonstrate the value of incorporating ResNum features into our evaluation framework, we consider ten promiscuous ligands and their reported PLIs from ChEMBL v33 (Zdrazil et al., 2024; Heinzke et al., 2024). We generate "positive" examples from the reported interactions and "negative" examples by recombining proteins and ligands from the positive set, excluding any pairs that are reported in ChEMBL. We compute misalignment scores for each example, scoring positives in a

leave-one-out manner by iteratively holding out each reported interaction. The ligands we consider target proteins with a broad range of functions; function-based evaluations are expected to differentiate poorly between positive and negative examples. We demonstrate that adding ResNum features can improve the separability of positive and negative examples. We also predict novel PLIs involving these ligands using PDBspheres (Zemla et al., 2022). We then evaluate the predictions with our framework, illustrating how the orthogonal information can be used to extract high-quality leads.

## 3 RESULTS AND DISCUSSION

Considering multiple modalities together can provide a more robust framework for identifying high-quality drug leads. For demonstration, we assemble sets of reported and unreported PLIs for promiscuous ligands and compare their misalignment-score distributions. Figure 1 plots these distributions and the associated ROC curves. As expected, the function-based misalignment scores do not clearly distinguish the two groups (AUROC = 0.603). Compared to the function-based scores, ResNum embeddings yield improved separation between the groups (AUROC = 0.763). Since negative examples include unreported interactions rather than experimentally validated non-binders, some true interactions may be labeled negative, limiting the attainable AUROC. By quantifying functional selectivity, Wakim et al. (2025) prioritize ligands that are strong candidates for function-based evaluations. By incorporating physicochemical representations, we broaden the set of ligands for which similar orthogonal evaluations are possible. These results suggest that adding additional orthogonal comparators, like ResNum, can improve the reliability of CADD.

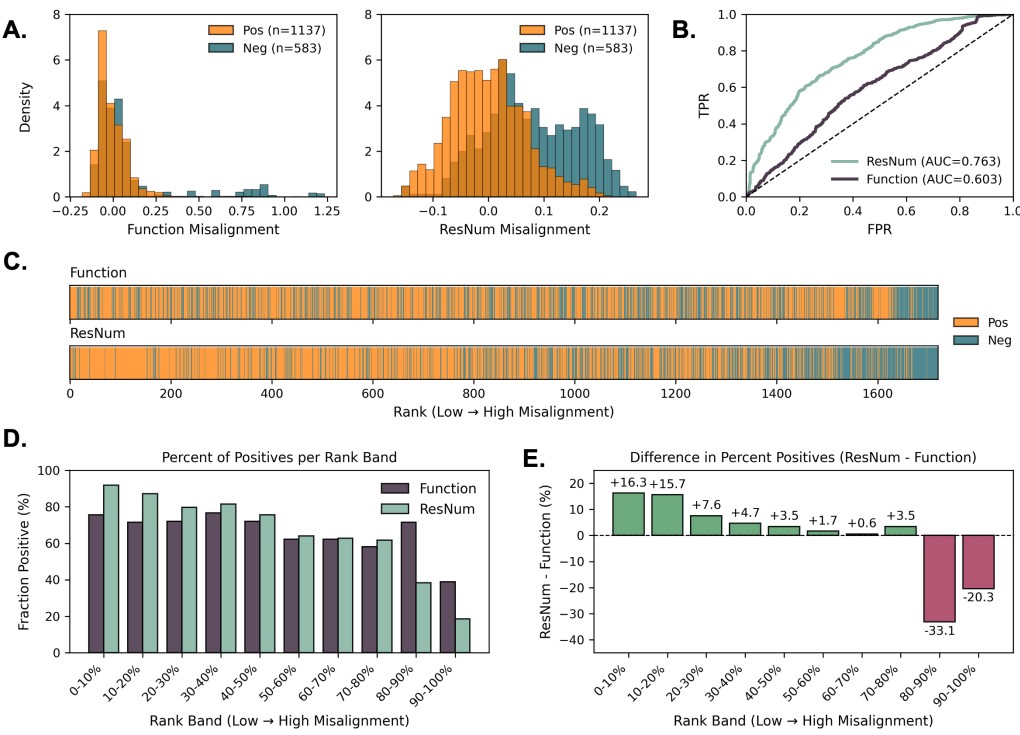

Figure 1: *Benchmarking functional and physicochemical evaluation metrics using reported (positive) and unreported (negative) PLIs involving promiscuous ligands. **(A)** Distributions of misalignment scores derived from (left) functional and (right) ResNum embeddings demonstrate greater separability based on physicochemical properties. **(B)** This enhanced separability is highlighted with ROC curves, which indicate a 0.16 improvement in AUROC. **(C)** By inspecting the rank positions of reported and unreported interactions, we see that ResNum embeddings better enrich for reported interactions at low misalignment scores. This is highlighted by **(D)** fractions of reported PLIs within bands of the ranked positions and **(E)** respective differences between ResNum- and function-based methods.*

We implement a multi-stage screening pipeline to illustrate how our protein representations can support drug discovery. First, using protein structural models and PDBspheres (Jumper et al., 2021; Varadi et al., 2022; Zemla et al., 2022), we predict 968 PLIs involving the ten promiscuous ligands from above. Among our predicted interactions, 745 (77.0%) are reported in ChEMBL v33 (Zdrazil et al., 2024; Heinzke et al., 2024). The remaining 223 (23.0%) interactions are not reported in the database. We filter these interactions based on functional and physicochemical misalignment scores, using a threshold of zero to distinguish high- and low-quality predictions. 84 (37.7%) of the 223 putative interactions may be excluded due to functional misalignment with reported PLIs, and an additional 103 (46.2%) may be excluded due to physicochemical misalignment. This leaves 36 (16.1%) remaining unreported interactions, which are supported by structural, functional, and physicochemical evidence. These interactions could offer leads for future therapeutic development. Combinations of predicted and known targets could offer clues into the pathways underlying a ligand's effects and toxicity.

One challenge in data-driven models for predicting PLIs is their dependency on negative examples, which are often underrepresented in experimental data. Given the mechanistic basis of our approach, our predictions are less biased by the underrepresentation of non-interacting protein-ligand pairs. Proteins in the putative interactions captured by our approach contain binding pockets that are structurally and physicochemically compatible with their predicted binding partners, and they share functional similarities with known targets of those ligands. However, while we reduce the reliance on negative examples, our approach hinges on representative annotations of protein functions and representative sets of drug targets (positive examples) reported in the literature. Our function and physicochemical evaluations also require that targets of ligands be sufficiently differentiated from random proteins. The aggregation of multiple independently measured modalities increases the likelihood of detecting a signal on which to differentiate high- and low-quality predictions.

## 4 Next Steps

We envision our structural, functional, and physicochemical representations as complementary components of a mechanistic framework for studies on drug repurposing and toxicity. Toward this vision, additional model development is needed to address questions such as: How should we fuse representations or aggregate misalignment scores to improve robustness? What thresholds should we apply when filtering PLIs based on misalignment scores? Should different thresholds/screening pipelines be applied based on ligand selectivity? Given the chemical diversity of small-molecule therapeutics, some physicochemical features may be more relevant for certain ligands and protein families than others. Accordingly, future development will also involve ligand-specific selection of ResNum features according to correlations observed in reported interactions. We are actively pursuing these directions, and a manuscript on this work is in preparation (Zemla et al., in preparation). We welcome input from the ICLR community. Using our approach, we aim to provide tools and resources that enhance the reliability of CADD, increase sensitivity for off-target binding, and improve the efficiency of therapeutic development. The framework presented here illustrates a key step toward this goal.

## Meaningfulness Statement

Interactions between small molecules and proteins drive therapeutic and off-target effects of drugs. In this work, we develop orthogonal representations that encode structural, functional, and physicochemical features of proteins and their binding pockets. These representations can be used during drug discovery to predict and evaluate PLIs. By incorporating biologically linked but independently derived representations, our work can improve the efficiency and reliability of CADD.

## Disclosure of LLM Usage

An LLM (ChatGPT, OpenAI) was used to polish the wording/grammar of the manuscript and assist with implementing basic utility functions and plotting code. The authors verified all content.

ACKNOWLEDGMENTS

This work has been authored by Lawrence Livermore National Security, LLC under Contract No. DE-AC52-07NA27344 with the U.S. Department of Energy. This material is based upon work supported by the Department of Energy, Office of Science, Office of Advanced Scientific Computing Research. The United States Government retains, and the publisher, by accepting the article for publication, acknowledges that the United States Government retains a non-exclusive, paid-up, irrevocable, worldwide license to publish or reproduce the published form of this work, or allow others to do so, for United States Government purposes. Release Number: LLNL-CONF-2015629.

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
