# OpenReview forum: "Orthogonal Evaluations Enable More Robust Predictions of Protein-Ligand Interactions"
_ICLR.cc/2026/Workshop/LMRL — ICLR 2026 Workshop LMRL Poster_

### Official Review · Reviewer_AbKL · 2026-02-24
**Novel ResNum Features for Promiscuous Ligands-Protein Interaction Lack Ablation and Modern Benchmarks**

**Rating:** 4
**Confidence:** 4

**Review:**

**Summary**
This paper addresses protein-ligand interaction (PLI) prediction with a specific focus on promiscuous ligands ,ie ligand that bind proteins with diverse or unrelated function. To do so, they introduce new pocket level features for the protein called **ResNum**  assigned based on similarity to known PLIs and local alignment to the target protein (known interacting protein).

While the motivation for moving beyond function-based representations is sound, the contribution feels incomplete: the feature design lacks justification, ablations are absent, and the benchmarking is insufficient to establish state-of-the-art relevance.

**Pros:**
* Their objective is really clear and defined, promiscuous ligands are acknowleged as an hard spot.
* The authors try to put forward physically-grounded/interaction-derived features. This is a principled design choice that could offer interpretability advantages over purely learned representations.

**Cons**
*The 235 physicochemical features are one of the paper's primary contributions, yet they receive only a brief summary. There is no justification for why these specific features were selected, nor a formal description sufficient for reproducibility. This is a significant omission.
*There is a clear lack of ablation, authors don't try with subseted set of features, not enabling us to comprehend from which features does the enhancement comes from, if all the features are mandatory or not. This would be a highly positive addition as their approach permits more interpretability in the protein-ligand behaviors.
*The only baseline is function similarity, which the authors themselves acknowledge is poorly suited to promiscuous ligands. Modern PLI methods both structure-based end-to-end approaches and affinity predictors, such as  should be included to provide meaningful and fair comparison for the improvements in performance claimed by the authors [1, 2, 3].

*References*
[1] Passaro et al., Boltz-2: Towards Accurate and Efficient Binding Affinity Prediction, 2025.
[2] Moon et al., PIGNet2: A Versatile Deep Learning-based Protein–Ligand Interaction Prediction Model for Binding Affinity Scoring and Virtual Screening, Digital Discovery, 2024.
[3] Jiang, S., et al. (2025). FlashAffinity: Bridging the Accuracy-Speed Gap in Protein-Ligand Binding Affinity Prediction. Machine Learning for Structural Biology Workshop, 2025.

---

### Official Review · Reviewer_YEAC · 2026-02-25
**Review: Orthogonal Evaluations Enable More Robust Predictions of Protein-Ligand Interactions**

**Rating:** 5
**Confidence:** 4

**Review:**

The paper proposes a multimodal framework to evaluate and filter predicted protein-ligand interactions (PLIs) by combining structural, functional, and physicochemical representations. The authors highlight a specific failure mode in current pipelines: function-based evaluations perform poorly for promiscuous ligands because these molecules target proteins with diverse and unrelated biological functions. To address this, the authors introduce a physicochemical comparator using "ResNum" features (capturing hydrophobicity, aromaticity, sulfur content, etc.) derived from AlphaFold 3 structural models and PDBspheres. In a preliminary evaluation involving 10 promiscuous ligands from ChEMBL, the addition of ResNum-based misalignment scores outperformed the function-only baseline.
The task could be better-motivated in the introduction and could use better clarification of where and when such methods would be applicable. The test set used for evaluation is quite small and the “negatives” are not true negatives but reshuffled positives which are known to have issues. Would it be possible to use experimentally verified inactive pairs (e.g., data from high-throughput screening assays in PubChem or specific inactive flags in ChEMBL)? This would better reflect a more real-world setting where the properties of proteins and ligands involved may also be different distribution-wise than for shuffled negatives.
While confidence estimation in CADD is a critical problem and the results presented here are promising, they are too preliminary and the sample size too small to confidently assess whether this framework actually solves it.

---

### Meta-Review · Area_Chair_4gc5 · 2026-02-28

**Recommendation:** Accept (Poster)
**Confidence:** 4

**Metareview:**

Accept.

---

### Decision · Program_Chairs · 2026-03-02

**Decision:**

Accept (Poster)

**Comment:**

Please see the meta-review.